# Changes in the Habitat Preference of Crested Ibis (*Nipponia nippon*) during a Period of Rapid Population Increase

**DOI:** 10.3390/ani11092626

**Published:** 2021-09-07

**Authors:** Liming Ma, Xinhai Li, Tianqing Zhai, Yazu Zhang, Kai Song, Marcel Holyoak, Yuehua Sun

**Affiliations:** 1School of Life Sciences, Hebei University, 180 Wusi East Road, Baoding 050024, China; maliming717@163.com; 2Key Laboratory of Animal Ecology and Conservation Biology, Institute of Zoology, Chinese Academy of Sciences, 1-5 Beichen West Road, Beijing 100101, China; songkai@ioz.ac.cn; 3University of Chinese Academy of Sciences, Yuquan Road, Beijing 100049, China; 4Shaanxi Hanzhong Crested Ibis National Nature Reserve, Hanzhong 723300, China; zhaitianqing@163.com (T.Z.); zx51w@126.com (Y.Z.); 5Department of Environmental Science and Policy, University of California, 1 Shields Avenue, Davis, CA 95616, USA; maholyoak@ucdavis.edu

**Keywords:** animal conservation, habitat preference, interaction effects, model selection, nest site, wetland

## Abstract

**Simple Summary:**

In 1981, the crested ibis (*Nipponia nippon*) was a critically endangered bird with only two pairs left in the world. Now, numbers have increased, and it has been reintroduced to many places in China, Japan, and Korea. In the breeding season, the crested ibis has high food needs (fish, freshwater invertebrate, etc.) to feed nestlings, and it prefers areas with rice paddies and waterbodies near to trees for nesting. We compared nest site preference from 1981 to 2019, and found that nest sites moved closer to houses of local farmers. In the 1980s, crested ibises only lived in remote mountain areas, where the farmers were too poor to afford pesticide and fertilizer, and consequently rice paddies contained substantial fish populations. In the meantime, the massive use of pesticides in more populated areas had caused the collapse of fish populations. A pesticide ban was enforced from 1981 on, when the crested ibis was rediscovered, and the freshwater biodiversity gradually recovered. The crested ibis returned to lower and more populated areas containing their ancestral habitats with large areas of wetlands.

**Abstract:**

The number of breeding pairs of crested ibis (*Nipponia nippon*) in Hanzhong, China has recovered remarkably from 2 to 511 from 1981 to 2019. Although the crested ibis has been closely monitored, the habitat preference of the bird has not been well studied despite the extensive increase in abundance. We used nest site data from the past 39 years and 30 environmental variables to develop species distribution models for each year. We applied random forest to select important environmental variables, and used logistic regressions to quantify the changes in habitat preferences in 39 years, taking into account the effects of interaction and quadratic terms. We found that six variables had strong impacts on nest site selection. The interaction term of rice paddies and waterbodies, and the quadratic term of precipitation of the wettest quarter of the year were the most important correlates of nest presence. Human impact at nest sites changed from low to high as birds increased their use of ancestral habitats with abundant rice paddies. We concluded that during the population recovery, the crested ibises retained their dependence on wetlands, yet moved from remote areas to populated rural regions where food resources had recovered due to the ban of pesticide use.

## 1. Introduction

Species can be described as occupying ecological niches [1] and having certain preferences for environmental conditions [2,3,4]. Species distribution models can quantify such preferences, and rank the importance of different environmental variables [5,6]. The habitat preferences of a species could change over time due to exogenous anthropogenic pressures (e.g., human disturbance) or environmental variation, or endogenous factors such as intraspecific competition. However, long-term studies are required to study such changes and these are extremely rare in the literature.

The crested ibis (*Nipponia nippon*) has experienced a dramatic increase in numbers [7,8]. Wild populations increased from two breeding pairs in the whole world in 1981 to 511 pairs in Hanzhong, central China in 2019. The habitat preferences of the crested ibis have been studied and key variables identified [8,9,10,11,12,13,14,15,16]. However, although the population density of the ibis has greatly changed, no study has compared the changes in habitat preference of the species. Doing so would help to inform the species’ management.

The crested ibis has been closely monitored for 39 years since its rediscovery in 1981 [17]. Its population fluctuated at a very low level in the first 16 years, with less than seven pairs [18], and then increased exponentially. Meanwhile, the local economy has grown rapidly in the area of occurrence in China, and conservation policies have been effectively implemented [19]. Both the ibis population and its surrounding environment have changed in the past four decades. In this study, we assessed and compared the habitat preference of the ibis through the bird’s recovery period. This enables us to make inferences about the reasons for population recovery.

## 2. Methods

### 2.1. Study Area

In 1981, the last two pairs of the crested ibis were found at Yang County, Hanzhong Prefecture, Shaanxi Province in central China [17]. Now, the crested ibis has expanded to neighboring counties in Hanzhong Basin [20], which is surrounded by the Qinling Mountains to the north and Daba Mountains to the south. The landscape comprises forested foothills and gently rolling croplands, mixed with rivers and ponds. Typical nesting habitat of the species consists of temperate forest mixed with rice paddies and other wetlands [17,21] (Figure 1).

### 2.2. Nest Site Data

The crested ibis was closely monitored since its rediscovery in 1981, and all nest sites were recorded and protected [19,22]. After 2006, the number of nests was over 100, and it was not easy to find all nests. Thereafter, local field investigators of the Shaanxi Hanzhong Crested Ibis National Nature Reserve conducted 3–4 line transect surveys every year to look for nests. They believe that all nests along the transect routes were located, since the nests of the crested ibis are large and visible, and the nesting areas are easily accessible, since the bird never lives far away from farmers [14]. We compiled the nest site data from 1981 to 2019. In 2020, the field surveys were suspended due to COVID-19 and no data were collected in that year.

### 2.3. Environmental Variables

We used 30 environmental variables to develop species distribution models for the crested ibis, including 19 climate variables [23], elevation [24], human footprint index [25] and land cover [26], as well as solar radiation, wind speed, and water vapor pressure for January and July, respectively [23] (Table 1).

Among the 30 variables, rice paddies and water bodies represent habitats containing food resources for the bird and are the most important habitats for nest site selection [27]. We used fine scale maps (1:100,000) to locate rice paddies and waterbodies [28]. The original polygon data of rice paddies and waterbodies were transformed to raster layers. Because the crested ibis usually fly a few hundred to a few thousand meters from their nests to reach foraging sites, the distance from nest sites to the forage sites is expected to be an important factor. The presence/absence of wetland was not found to be a good indicator for nest site selection. We estimated the values of probability of presence of rice paddies and waterbodies from binary (0/1) values by predicting the presence of wetlands using random forest [29] with longitude and latitude as explanatory variables (Appendix A). All of the 30 variables were raster layers covering the study area (Figure 1) with the resolution of 1 km.

The crested ibis always selects tall trees for nesting. Most nest trees are oak (*Cyclobalanopsis glauca*) and larch (*Larix gmelinii*). The 30 environmental variables represent the habitat characteristics at nest tree locations. Distance to rivers is not a key factor, nor the distance to rice paddies or lakes. Nest trees were generally located near a wetland, or a few kilometers away from any wetlands.

### 2.4. Selecting Important Variables

We used random forest [29] to select the key variables among the 30 environmental variables predicting nest presence/absence. The variables might have complex high order and/or interaction effects on nest site selection. Random forest is robust for highly dimensional data and complex nonlinear effects [30,31]. Two indices for variable importance were provided by the randomForest package [32] in the R language [33]. Gini importance was reported to suffer from the bias of the underlying Gini-gain splitting criterion, while the alternative permuted MSE (mean squared error) importance is generally accepted as a reliable measure of variable importance [34]. As such, we used increased MSE when a variable was removed from the model to weight the 30 environmental variables.

We used 900 evenly distributed points as pseudo-absence data, and the nest sites as presence data. We built a model for each year, so that we had 39 models for years from 1981 to 2019. The environmental variables were ranked in each model.

### 2.5. Checking Quadratic and Interaction Terms

After determining important variables for nest presence/absence, we checked the contribution of quadratic and interaction terms. A quadratic term implies an optimum value for the species, which usually fits environmental variables such as temperature, precipitation, and elevation [35]. Interaction terms are also common in species distribution models [27]. We applied logistic regressions to check the contribution of all two-way interaction terms between each pair of the important variables, and all quadratic terms. We demonstrated the effects of each variable using the partial plot function provided by the randomForest package to show whether the quadratic terms or high order terms should be considered.

### 2.6. Contribution of Variables and Terms

We applied dominance analysis to determine variable importance and the contribution of interaction and quadratic terms in logistics regressions [36]. McFadden R^2^ values were calculated to quantify variables’ contributions using the R package dominanceanalysis [37].

## 3. Results

From 1981 to 2019, the crested ibis population increased from two breeding pairs to 511 breeding pairs (Figure 2), expanding from Yang County to other neighboring counties in Hanzhong Basin (Figure 1).

We ranked the importance of the 30 environmental variables in nest site selection from 1981 to 2019, and found the top six variables were as follows: solar radiation in January, precipitation during the wettest quarter (WorldClim variable 16), precipitation during the warmest quarter (WorldClim variable 18), probability of waterbody presence, probability of rice paddy presence, and human footprint index (Figure 3). The values of the six variables at nest sites are quite different from those at 900 pseudo-absence sites (Appendix A).

The six important variables are independent, except that precipitation of wettest quarter and precipitation of warmest quarter are almost identical. We removed precipitation of warmest quarter, and the variance inflation factor (VIF) of the five remaining variables was below 10 (Table 2). VIF is a measure of variable independence in a dataset.

We built logistics regressions to analyze nest site selection for each year from 1981 to 2019. The regression models included five linear terms, ten two-way interaction terms, and five quadratic terms of the five important variables. The average deviance of each variable and term across 39 logistic regressions for years from 1981 to 2019 are listed in Table 3. The interaction term of rice paddy and waterbody was the most important term, and the quadratic term of precipitation of wettest quarter ranked second.

The partial plots of the rice paddy (Appendix A), waterbody (Appendix A), and solar radiation (Appendix A) show monotonous positive effects, indicating that the crested ibis preferred areas with high density of rice paddies and waterbodies, and high solar radiation. The quadratic curves of the partial plots for precipitation during the wettest quarter (Appendix A) and precipitation during the warmest quarter indicate the bird needed an optimum value of the variables. The partial plots of human footprint index changed from 1981 to 2019 (Figure 4). In the early years before 2004, the crested ibis nested at remote areas with low human footprint index. From 2004 to 2009, the patterns were irregular. From 2010 to 2015, the bird preferred areas with higher human footprint index. From 2016 to 2019, the bird mostly nested in areas with an intermediate human footprint index (Figure 4).

We fitted a single model to represent the probability of nest presence for all years from 1981 to 2019. This model allowed us to quantify the contribution of variables and terms across years. Rice paddy, waterbody, and solar radiation had consistent monotonous positive effects on nest site selection, so that their linear terms were used in the model. The interaction term of probability of presence of rice paddy and waterbody and the quadratic term of precipitation during the wettest quarter were the two top-ranked terms (Table 3), and they were added into the model. The effect of human footprint index was complex (Figure 4) and it was added to the model as a linear term. The model was
logit(p) ~ rice_paddy + waterbody + solar1 + rice_paddy:waterbody + bio_16 ^2^ + footprint.

We applied dominant analysis to quantify the contribution (R^2^) of each term in this model (Figure 5). The interaction term of probability of rice paddy presence and probability of waterbody presence had the highest contribution, especially in the period from 1990 to 2010.

## 4. Discussion

The wild population of crested ibis increased from two breeding pairs to 511 pairs in the Hanzhong Basin from 1981 to 2019, and the species’ habitat preferences changed during this period. In the past four decades, the distribution center of the crested ibis moved from remote mountains to lower and more populated areas with large areas of wetlands [18]. Our analyses show that the bird lived far away from human residences before 2004 and moved towards areas with increased human footprint from 2004 to 2015, and has stayed at areas with an intermediate human activity level since 2016 (Figure 4). This represents a shift towards nesting in the trees around farm residences. Such changes were not due to adaptation, but because habitat conditions have improved [9,14,15,38]. A ban on pesticide use and poaching control starting from 1981 gradually improved the habitat quality so that the crested ibis returned to their ancestral (pre-1980s) breeding sites in lower altitude areas [9,14,20].

Habitat selection is a complicated behavior and is usually associated with many factors [39,40]. In this study, we compiled 30 environmental variables (Table 1). Traditional logistic regression cannot handle as many as 30 variables because multicollinearity would be severe. Random forest is an ideal algorithm for dealing with such highly dimensional data [30]. However, random forest has a complex structure and is not a transparent model. After selecting important variables using random forest, we applied logistic regressions to quantify the contribution of linear, quadratic, and interaction terms to explaining nest presence/absence. The interaction and quadratic terms ranked higher than the linear terms for nest site selection of the crested ibis (Table 3).

In this study, the most important term is the interaction of the probabilities of presence of rice paddies and waterbodies, which is consistent with the study using watersheds (not occurrences) as observations (rows in the dataset) [27]. The interaction means the bird needs both wetland types, which cannot be compensated for by each other. If rice paddies can be replaced by waterbodies, the sum of the two variables, not the product, would be important. The crested ibis needs rice paddies and waterbodies at different periods, and both of them must be above certain minimum areas (Appendix A). The importance of rice paddies and waterbodies indicates that food resources are key and limited factors for habitat selection by the crested ibis, with climate conditions and human impacts ranking lower in importance (Table 3 and Figure 5).

The quadratic term of precipitation during the wettest quarter (WorldClim variable bio_16) ranked second (Table 3), implying that the crested ibis needs a certain value of precipitation around 400 mm in the three summer months in the wettest quarter (Appendix A). Higher or lower precipitation made the nesting habitat less suitable.

In the past century, the crested ibis population declined severely in East Asia [14,41]. In the meantime, a related species, the black-headed ibis (*Threskiornis melanocephalus*), and sympatric little egrets (*Egretta garzetta*) remained stable [42,43]. The blacked-headed ibis also occurs in East Asia. It builds nests on the ground, and its breeding areas in Northeast China are remote and flat marshes with a low human activity level. The crested ibis needs tall trees for nesting. Wetlands mixed with forests are suitable for humans, and all original habitats of the crested ibis had been developed for agriculture [14,44]. The crested ibis has adapted to farmer-dominated wetlands, and has lived with farmers for hundreds of years. Unfortunately, the chemical revolution in agriculture resulted in the population collapse of the crested ibis [14]. The little egret also experienced the massive use of pesticide. However, unlike the crested ibis, it does not depend on rice paddies. The little egret feeds extensively on fishes in rivers, aided by sharp eyesight and a straight sharp bill. The crested ibis, with its down-curved bill, feeds by probing the bottom of rice paddies and detects the movement of fishes. Such a forage strategy makes the crested ibises depends on soft-bottom wetlands such as rice paddies, which were contaminated by pesticides. Fish populations in rivers were less influenced by pesticides, so that little egret populations remained stable.

Among the recovered species of the world, a few species were as closely monitored with detailed demographic data as the crested ibis, such as the Mauritius kestrel [45], the black-footed ferret [46], wolves on the Scandinavian peninsula [47], Hawaiian green sea turtles [48], the California condor [49], a re-introduced population of wild dogs [50], and Virunga mountain gorillas [51]. Yet, the use of spatially explicit population information for recovering species is rare [52]. Using accurate nest location data over the past 39 years, we were able to analyze the dynamics of habitat preference (Figure 4 and Appendix A), showing the changes in the contribution of key factors (Figure 5) for nest site selection of the crested ibis.

Small populations usually suffer from the bottleneck effect and inbreeding depression [53,54]. The crested ibis, recovered from two pairs, experienced extremely high inbreeding, and resulted in very low genetic diversity [55]. In fact, most individuals are offspring of a single pair of “super” parents that nested in Sanchahe from 1984 at elevation 1200 m. So far, the phenotype, including traits such as color, size, shape, and behavior, is normal. The crested ibis even reached 7000 individuals in 2021, but might still be susceptible to infectious diseases that could cause the population to collapse.

## 5. Conclusions

Habitat selection of animals is a complex process involving many environmental factors. We used a long-term nest site dataset of an endangered bird, compared 30 environmental variables and filtered unimportant factors, quantified interaction and quadratic terms, and found meaningful effects of some variables. We concluded that the crested ibis’ preference for rice paddies and waterbodies, together with high solar radiation, remained the same from 1981 to 2019. The preference for certain precipitation levels remained constant, yet its preference for human impact changed substantially, because the habitat value of rice paddies was recovered due to a pesticide ban in 1981. This likely caused an increase in the food resources in rice paddies.

## Figures and Tables

**Figure 1 animals-11-02626-f001:**
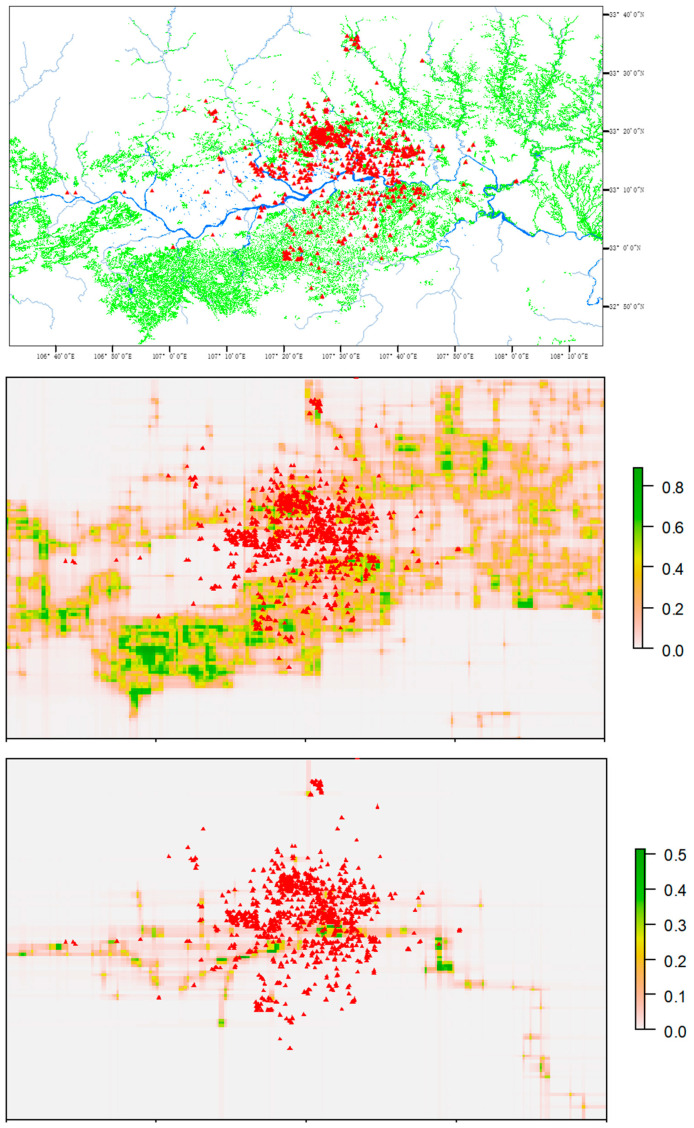
Distribution of nest sites (red triangles) of the crested ibis in 2019 in Hanzhong Basin, Shaanxi Province, China. The background is an elevation map. The bright green areas are rice paddies, and the blue areas are rivers, reservoirs, and lakes. The brown lines are roads. The dark brown area by the road is the human residence area of the city Hanzhong.

**Figure 2 animals-11-02626-f002:**
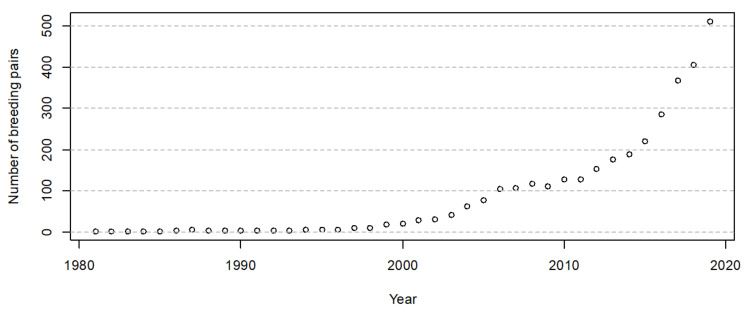
The number of breeding pairs of the crested ibis in Hanzhong Basin from 1981 to 2019.

**Figure 3 animals-11-02626-f003:**
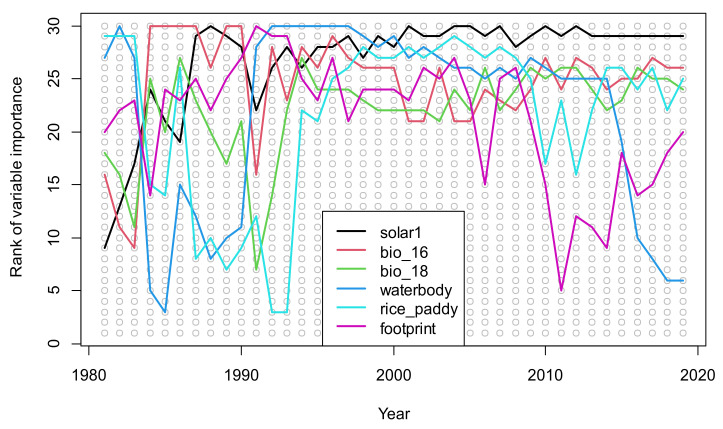
The ranks of top-ranked environmental variables for the nest site selection of the crested ibis from 1981 to 2019 using the index %IncMSE in random forest. Rank 30 is the highest rank. The top six variables were solar radiation in January, precipitation of the wettest quarter (WorldClim variable bio_16), precipitation of the warmest quarter (WorldClim variable bio_18), probability of waterbody presence, probability of rice paddy presence, and human footprint index.

**Figure 4 animals-11-02626-f004:**
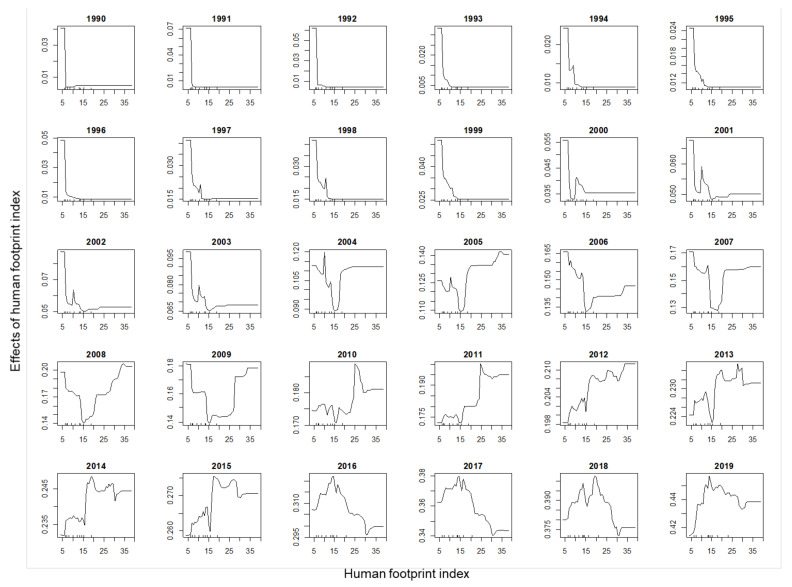
The partial plots showing the effects of human footprint index on nest site selection of the crested ibis from 1990 to 2019 based on random forest.

**Figure 5 animals-11-02626-f005:**
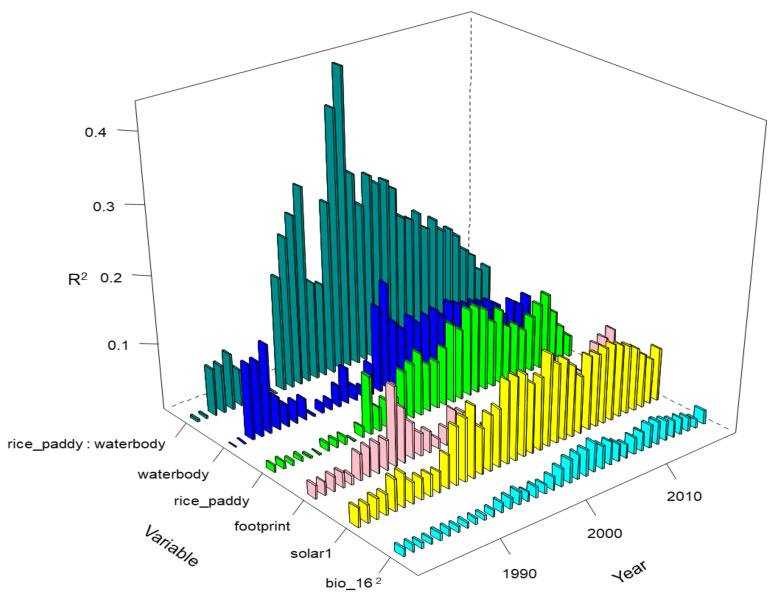
The R^2^ values of the most important variables and terms for the nest site selection of the crested ibis from 1981 to 2019.

**Table 1 animals-11-02626-t001:** Variable parameters and data sources used in the analysis of nest site selection of the crested ibis. The variable names used in the models are in parentheses.

Variables	Parameters				Unit	Citation
	Mean	Minimum	Maximum	SD		
Nineteen variables for temperature and precipitation (bio_1 to bio_19)	/	/	/	/	/	[23]
Elevation (elev)	1091.3	354	2572	472.7	M	[24]
Human footprint index (footprint)	11.38	4.00	43.12	5.63	/	[25]
Land cover (landcover)	/	/	/	/	/	[26]
Rice paddies (rice_paddy) ^1^	0.107	0	0.888	0.156	square km	Predicted values
Waterbodies (waterbody) ^1^	0.008	0	0.513	0.034	square km	Predicted values
Solar radiation in January (solar1)	8516	7733	9203	288.9	kJ m^−2^ day^−1^	[23]
Solar radiation in July (solar7)	19,439.7	18,020	20,249	444.8	kJ m^−2^ day^−1^	[23]
Wind speed in January (wind1)	1.718	1.1	2.9	0.283	m s^−1^	[23]
Wind speed in July (wind7)	1.673	1.3	2.8	0.245	m s^−1^	[23]
Water vapor pressure in January (vapor1)	0.432	0.25	0.57	0.072	kPa	[23]
Water vapor pressure in July (vapor7)	2.172	1.46	2.65	0.264	kPa	[23]

^1^ Predicted values representing the probability of wetland presence based on land cover data [28].

**Table 2 animals-11-02626-t002:** Variance inflation factors (VIFs) for the five variables that were important for nest site selection of the crested ibis from 1981 to 2019 after precipitation of the warmest quarter (bio_18) was removed from the model.

Bio_16	Solar1	Footprint	Rice_Paddy	Waterbody
5.846	6.023	2.861	1.349	2.610

**Table 3 animals-11-02626-t003:** The average deviance of each variable and term in 39 logistic regressions for nest site selection of the crested ibis from 1981 to 2019. In the table, colon indicates an interaction term (R style).

Variable/Term	Deviance
rice_paddy:waterbody	79.74
I(bio_16^2)	67.41
solar1:waterbody	55.05
bio_16:solar1	46.02
solar1	36.43
solar1:rice_paddy	34.20
rice_paddy	29.33
footprint:waterbody	29.24
solar1:footprint	24.82
footprint	24.20
waterbody	18.80
bio_16:waterbody	17.01
footprint:rice_paddy	15.31
bio_16:footprint	14.31
I(footprint^2)	12.20
I(solar1^2)	9.26
I(waterbody^2)	9.05
bio_16	8.94
I(rice_paddy^2)	8.45
bio_16:rice_paddy	3.04

## Data Availability

These data are available upon request from the corresponding author.

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
