# Peer review of "Changes in the Habitat Preference of Crested Ibis (Nipponia nippon) during a Period of Rapid Population Increase"

_animals, 2021, doi:10.3390/ani11092626_

Round 1

Reviewer 1 Report

General comment 

This manuscrip reports the changes of habitat preference of the crested ibis which is an extremely small population and  is recovering now in its traditional distribution area. The results will  is helpful for us to understand the recovery process of the endangered species and evaluate the protection effectiveness. 

The authors should pay attention about the nest tree and river variables, that is important to nest site selection.

Specific comments:

1. Line 18-19, "In 1980s the crested ibis only lived in remote mountain areas, where the farmers were too poor to afford pesticide and fertilizer."  About poor I hope to put it another way, Different models of agricultural development have been adopted in different historical periods.

2. Line 77, about the evironmental variables, nest tree and river are important nest choice factors, it should be paid attention.

3.Line 94-96, in Figure1, Line 96, In Figure 1, there are too manay dark brown, all the dark brown areas are human residence ? A legend in this picture  could make the figure more intuitional.

4. Line 98-99, a total of 30 environmental variables to develop species distribution models for the crested ibis. Why to choose these variables? Did Human footprint index include the population dense and traffic intensity, etc.

5. In Figure 4, the meaning of axes are missing. Add title of axes.

6. In Figure 5, The histogram of variables covers each other. Please change one direction to avoid this problem

7. In 5. Conclusions, some content can be put into the discussion。 

Author Response

General comment 

This manuscript reports the changes of habitat preference of the crested ibis which is an extremely small population and  is recovering now in its traditional distribution area. The results will be helpful for us to understand the recovery process of the endangered species and evaluate the protection effectiveness. 

Reply: Thank you very much for your pertinent and constructive comments.

The authors should pay attention about the nest tree and river variables, that is important to nest site selection.

Reply: Yes. The crested ibis always selects tall trees for nesting. Most nest trees are oak (Cyclobalanopsis glauca) or pine (Larix gmelinii). Distance to rivers is not a key factor, nor the distance to rice paddies or lakes. The nest trees can be near a wetland, or a few kilometers away from wetlands. We added above contents to the manuscript (lines 94-98).

Specific comments:

  1. Line 18-19, "In 1980s the crested ibis only lived in remote mountain areas, where the farmers were too poor to afford pesticide and fertilizer."  About poor I hope to put it another way, Different models of agricultural development have been adopted in different historical periods.

Reply: We understand the reviewer’s point that several decades ago farmers looked poor in a “modern” eye. We mean poor here by comparing with other regions at the same time. I used to live in their houses in 1990s, and they really had nothing, no furnisher, no electronic power, even no quilt. At the same period, farmers in lower area have basic living facilities.

  1. Line 77, about the evironmental variables, nest tree and river are important nest choice factors, it should be paid attention.

Reply: Nest tree is a fine scale variable. We had analyzed the importance of fine scale variables such as height of nest tree, nest position (on south side of the trunk or north side of the trunk), coverage above the nest, nest tree position (valley bottom, or middle of the mountain, or top of the mountain), tree density around nest tree, etc. (Li et al. 2001). In this study, we analyze the nest site selection at a larger spatial scale, focus on regional characteristics, so that we did not use variables about nest trees. River is a part of wetland system. We used two types of wetlands: rice paddy and waterbody. Rivers, ponds, lakes are waterbodies. We clarified this point in the manuscript.

Li, X. H., Z. J. Ma, D. M. Li, C. Q. Ding, T. Q. Zhai, and B. Z. Lu. (2001) Using resource selection functions to study nest site selection of crested ibis. Biodiversity Science, 9, 352-358.

3.Line 94-96, in Figure1, Line 96, In Figure 1, there are too manay dark brown, all the dark brown areas are human residence ? A legend in this picture  could make the figure more intuitional.

Reply: We agree our description is confusing. The large dark brown areas are high elevated areas. The small dark brown areas in Hanzhong Basin are human residences. To be specific, we changed the sentence in legend to “The dark brown area along the road is the human residence area of the city Hanzhong.”

  1. Line 98-99, a total of 30 environmental variables to develop species distribution models for the crested ibis. Why to choose these variables? Did Human footprint index include the population dense and traffic intensity, etc.

Reply: Because the 30 variables are important and accessible. Our previous studies had proved rice paddy and water body are most important factors. Human footprint index includes four features: population density, land transformation, human access (i.e. reads), and power infrastructure.

  1. In Figure 4, the meaning of axes are missing. Add title of axes.

Reply: We added the title for Y axes as:Effects of human footprint index

  1. In Figure 5, The histogram of variables covers each other. Please change one direction to avoid this problem

Reply: We had adjusted the aspect of this 3D plot. However, it is inevitable that some bars are blocked. It is not necessary to show all details in this figure.

  1. In 5. Conclusions, some content can be put into the discussion。

Reply: Agree. We rewrote the conclusion section, moved the statements about the cases of other species to the discussion section (lines 218-242).

Reviewer 2 Report

The authors present a long-term study of 39 years of 30 environmental parameters that possibly influenced habitat choice in crested ibis in China. I find the study well planned, conducted, analysis appropriate, and the discussion supports the findings. This is an important paper for a recovering species. However, the paper must be proofread and edited by a native English speaker for the flow of language and correction of grammatical mistakes.  Some examples are given here but the whole paper needs to be edited properly -

Pg 1, 2 -  “endogenous factors (e.g. competition), ….”  Is competition endogenous? It is an external influence. Change the example to something more appropriate (internal).

Pg 2, line 55 – “ …. and conservation policies have been in effect …” change to implemented

Pg 2, line 71 – “Local stuff of the ….”  STUFF????? This is incorrect usage of language in a  scientific paper.

Further, it is of interest that the species is recovering but I think the authors should consider relating to the genetic bottleneck of the species and that could influence habitat selection.

Also, the discussion is very brief and no comparisons with any other species were attempted. The authors should compare their findings to the species in other habitats/studies published, sister species, other wetland/paddy species? This subject is dismissed in one sentence in conclusions and that’s wrong. Comparisons do not have to be only with recovering species forced to live in anthropogenic-modified habitats.

Author Response

The authors present a long-term study of 39 years of 30 environmental parameters that possibly influenced habitat choice in crested ibis in China. I find the study well planned, conducted, analysis appropriate, and the discussion supports the findings. This is an important paper for a recovering species. However, the paper must be proofread and edited by a native English speaker for the flow of language and correction of grammatical mistakes.  Some examples are given here but the whole paper needs to be edited properly –

Reply: Thank you very much for your professional and constructive comments. As to language problems, we went through the manuscript many times, and corrected about 50 grammar or wording errors.

Pg 1, 2 -  “endogenous factors (e.g. competition), ….”  Is competition endogenous? It is an external influence. Change the example to something more appropriate (internal).

Reply: We mean competition among different individuals, i.e. within-species competition. We changed the sentence to (e.g. competition among individuals).

Pg 2, line 55 – “ …. and conservation policies have been in effect …” change to implemented

Reply: We changed the sentence to “…have been implemented and made changes”.

Pg 2, line 71 – “Local stuff of the ….”  STUFF????? This is incorrect usage of language in a  scientific paper.

Reply: We made a type error here. We mean “staff”, not “stuff”. We changed the sentence to “Local field investigators of the…”

Further, it is of interest that the species is recovering but I think the authors should consider relating to the genetic bottleneck of the species and that could influence habitat selection.

Reply: The crested ibis, recovered from two pairs, experienced extremely high inbreeding, and resulted in very low genetic diversity. In fact, most individuals are offspring of a single pair of “super” parents nested in Sanchahe from 1984 at elevation 1200 m. So far, the phenotype including traits such as color, size, shape, and behavior is normal. We added one paragraph about it in the discussions section (lines 249-255).

Also, the discussion is very brief and no comparisons with any other species were attempted. The authors should compare their findings to the species in other habitats/studies published, sister species, other wetland/paddy species? This subject is dismissed in one sentence in conclusions and that’s wrong. Comparisons do not have to be only with recovering species forced to live in anthropogenic-modified habitats.

We added one paragraph and compared the crested ibis with its sympatric species, little egret, and its sister species, the black-headed ibis (lines 224-240). The paragraph is: In the past century, the crested ibis population declined severely in East Asia [14, 41]. In the meantime, its relative species, the black-headed ibis (Threskiornis melanocephalus), and its sympatric species, little egret (Egretta garzetta), remained stable [42, 43]. The blacked-headed ibis also occurs in East Asia. It builds nests on the ground, and its breeding areas at Northeast China are remote and flat marshes, where has a low human activity level. The crested ibis needs tall trees for nesting. Wetlands mixed with forests are suitable for human, and all original habitats of the crested ibis had been developed for agriculture [14, 44]. The crested ibis has adapted to farmer-dominated wetlands, and lives with farmers for at least hundreds of years; unfortunately, chemical revolution in agriculture resulted in the population collapse of the crested ibis [14]. The little egret also experienced massive use of pesticide. However, unlike the crested ibis, it does not depend on rice paddies. The little egret has sharp eye sight and a straight bill, and it can see fishes and catch them. The crested ibis has a down-curved bill, and it usually repeatedly probes the bottom of rice paddies and detects the movement of fishes. Such a forage strategy makes the crested ibises depends on soft bottom wetlands such as rice paddies, where is contaminated by pesticide. Fish populations in rivers are less influenced by pesticide, so that the little egret remains stable.

We moved the contents about other recovered species from the conclusion section to the discussion section.

Round 2

Reviewer 2 Report

The authors have made a commendable effort to correct as per the comments of the reviewers. The paper is now much improved and almost ready. However, in spite of the comment that they have made more than 50 editorial corrections, language remains a problem. The authors need to get professional help, either through a linguistic agency or a computer program that corrects English. 

For example, page 2 line 46: "(e.g. competition among individuals)" - should be written (e.g. intra-specific competition). These seemingly minor, but professional, terms make all the difference in reading a scientific paper.

Good luck! 

Author Response

Thanks a lot for providing your comments so promptly. We had accepted the comment raised as an example at line 46. We went through the manuscript twice and made about 20 changes. If the current version does not meet the journal standard, we will look for a native English speaker or a commercial company to help us.